# The Influence of Ce, La, and SiC Particles Addition on the Formability of an Al-Si-Cu-Mg-Fe SiCp-MMC

**DOI:** 10.3390/ma15113789

**Published:** 2022-05-26

**Authors:** Andong Du, Lucia Lattanzi, Anders E. W. Jarfors, Jie Zhou, Jinchuan Zheng, Kaikun Wang, Gegang Yu

**Affiliations:** 1Department of Materials Processing and Control Engineering, School of Materials Science and Engineering, University of Science and Technology Beijing, Xueyuan Road 30, Haidian District, Beijing 100083, China; b20170186@xs.ustb.edu.cn; 2Institute of Semi-Solid Metal Technology, China Academy of Machinery Sciences and Technology (Jiangle), Huancheng East Road 22, Jiangle County, Sanming 353300, China; zhengjc@cam.com.cn (J.Z.); yugg@cam.com.cn (G.Y.); 3School of Engineering, Materials and Manufacturing, Jönköping University, P.O. Box 1026, 551 11 Jönköping, Sweden; lucia.lattanzi@ju.se; 4Jiangsu University of Technology, Zhongwu Road 1801, Changzhou 213001, China; zhou_j09@163.com

**Keywords:** aluminium, metal matrix composite, brake disk, thermal stability, forming, processing map

## Abstract

Road transport and the associated fuel consumption plays a primary role in emissions. Weight reduction is critical to reaching the targeted reduction of 34% in 2025. Weight reduction in moving parts, such as pistons and brake disc rotors, provide a high-impact route to achieve this goal. The current study aims to investigate the formability of Al–Si alloys reinforced with different fractions and different sizes of SiCp to create an efficient and lightweight Al-MMC brake disk. Lanthanum (La) and cerium (Ce) were added to strengthen the aluminium matrix alloy and to improve the capability of the Al-MMC brake discs to withstand elevated temperature conditions, such as more extended braking periods. La and Ce formed intermetallic phases that further strengthened the composite. The analysis showed the processability and thermal stability of the different material’s combinations: increased particle sizes and broader size range mixture supported the formation of the SiCp particle interactions, acting as an internal scaffolding. In conclusion, the additions of Ce and La strengthened the softer matrix regions and resulted in a doubled compression peak strength of the material without affecting the formability, as demonstrated by the processing maps.

## 1. Introduction

The International Energy Agency declared transport responsible for 25% of fuel consumption, and road transport plays a primary role in emissions. By 2025, transport emissions should be reduced by 34% to reach the goal that the global temperature will not increase by 2 °C [1]. Considering the electricity generation and materials production emissions, the weight reduction in automotive components has an immediate effect on emissions saving. Weight reduction is critical for moving parts, which have the most significant influence on energy consumption, and it could improve the vehicle’s driveability during acceleration and deceleration [2]. Al-based metal matrix composites (Al-MMCs) are valid candidates to substitute cast iron in the lightweight automotive application as pistons and brake discs [3,4].

After an initial interest in the 1990s [5,6], Al-based matrix composites have attracted attention again in recent years [7,8,9] due to their high specific modulus and specific strength, excellent wear resistance, and good stiffness. Typical components, such as pistons and brake discs, demand high strength even at high temperatures, which is the weak point of Al alloys. Several Al-based MMCs with different ceramic reinforcements find extensive application in the automotive industry, as summarised in Table 1.

The addition of transition metals [18,19,20] and rare earth elements [21,22] to the matrix alloy helps to improve the maximum operating temperature of the composite material. Several methods produce metal matrix composites [23], such as powder metallurgy, pressure infiltration, spray deposition, and stir casting. The least expensive method is stir-casting, and the stirring force improves the wettability between the particles and the molten metal [24]. Workability is another critical drawback of Al-based composite materials. The challenge lies in the difficulty of machining composite materials with high SiCp fractions due to the hardness of SiC [25]. Yamagata et al. [3] studied a new approach to manufacturing pistons and concluded that casting and forging offer an efficient route to near-net-shape processing, saving time, and improving production efficiency.

The Arrhenius equation has been widely used to describe the relationship between strain rate, flow stress, and temperature. The effects of temperature and strain rate on the deformation behaviour are represented by the Zener–Hollomon parameter (Z) in an exponent type equation [26]. Other work on Al-based composites with Zener–Hollomon includes Hao et al. [27] who studied the 35% SiCp/2024Al metal matrix composites and indicated that the flow stress behaviour of composite during hot compression deformation can be represented by a Zener–Hollomon parameter in the hyperbolic sine form. Lattanzi et al. [20] did research on the relationship between the Z parameter and temperatures and strain rates, and it indicated that peak stress at high temperatures and low strain rates was reduced because of dynamic recovery and recrystallisation, and Z values decreased. Conversely, the Z values increased at low temperatures and high strain rates, indicating dislocation generation. This phenomenon led to work hardening and higher peak stress. Patel et al. [28] studied the AA2014–10 wt.% SiCp composites; the Z parameter described the flow behaviour of the samples. The Z value decreased with increasing temperature, and it is essentially due to extensive dynamic softening.

The processing maps are an additional tool to analyse the hot compression response of the material and guide the choice of the hot working parameters as they show safe and unsafe ranges of temperatures and strain rates to avoid operational regions in which damage occurs. The extreme damage mechanisms were identified as cavity formation at hard particles in a soft matrix, occurring at low temperatures and high strain rates, and wedge cracking at grain boundaries, occurring at high temperatures and low strain rates [29]. Several works in the literature defined the processing map for Al-based composites, mainly Al–Cu systems. Huang et al. [30] investigated the hot deformation of an Al–Cu–Mg alloy reinforced with 14 vol.% of SiCp. The temperature range was 355–495 °C, and the strain rate range was 0.001–1 s^−1^. The authors assigned different deformation mechanisms to different temperature ranges and concluded that temperature had a more significant influence than strain rate on the material response. Hao et al. [27] investigated the deformation behaviour of an Al–Cu–Mg alloy reinforced with 35 vol.% of SiCp; according to the processing map and the micrograph, the authors concluded that the optimal workability temperature and strain rate were 500 °C and 0.1 s^−1^, respectively. Xiao et al. [31] investigated an Al–Cu alloy reinforced with 15 vol.% of SiCp produced by powder metallurgy. The highest efficiency value was at 500 °C and 0.001 s^−1^, a lower value than the results by Hao et al. [27]. Patel et al. [28] investigated the hot deformation behaviour of an Al–Cu system reinforced with 10 wt.% of SiCp. The authors reported that the failure mechanisms mainly involved interfacial cracking between particles/matrix and intermetallic/matrix. Decohesion was severe at low strain rates. Ramanathan et al. [32] studied the workability of an Al–Cu alloy reinforced with 15 vol.% of SiC particles. They reported that the optimum domain for dynamic recrystallisation occurred in the temperature and strain rate range of 360–460 °C and 0.1–0.7 s^−1^. Wedge cracking was observed in the temperature range 460–500 °C under a lower strain rate.

The current research gap is on the role of different fractions and sizes of the reinforcement, and the role played by the matrix alloy in the overall response of the composite material. In light of this literature analysis, the present study investigates the formability of Al–Si alloys reinforced with different fractions and different sizes of SiCp. Besides, to improve the high-temperature performance of the composite, lanthanum (La) and cerium (Ce) were added to the matrix alloy in one case. Previous works on La and Ce added to Al alloys demonstrated that, from a sustainability standpoint and business perspective, these elements support efficient strength, save money, and are environment-friendly [33]. The processing maps of the materials shed light on their formability and the role played by the size and fraction of the reinforcement [33].

## 2. Materials and Methods

This section describes the experimental procedure of the presented work and Figure A1 in Appendix A summarises it in a schematic representation.

### 2.1. Material Production

Five different SiCp reinforced Al-based composites were produced with different matrixes and sizes of carbide particles, and the composition is listed in Table 1. An amount of 20 wt.% of SiCp were added into the matrix, and the materials were processed by a proprietary stir-casting method to keep porosity at a minimum level. The carbides were heat-treated at 1000 °C for one hour to develop a layer of silicon oxide (SiO_2_) on the surface of particles. This treatment enables evenly dispersed SiC particles in the molten material because the wetting angle between SiO_2_ and molten Al will be below 68.8 degrees [34]. The C0 matrix alloy is the base alloy; the C1 alloy was obtained by adding the master alloys Al–30% Ce, Al–30% La, Al–50% copper (Cu), Al–10% nickel (Ni), and Al–20% manganese (Mn) to the C0 alloy. The 23 µm, 50 µm, 10 µm and, mix size SiCp were used in this research, and the dimension is the cut-off limit for the batch of particles. The compositions of the alloys were measured using direct current plasma emission spectroscopy (DCPMS) (ATI Wah Chang, Albany, OR, USA). Please note that due to analysis limitations Ce and La could not be assessed using DCPMS, and Ce and La are given as nominal values. The detailed information is shown in Table 2. Five different SiCp-reinforced Al-based composites were produced with different matrixes and sizes of carbide particles, and the composition is listed in Table 2.

### 2.2. Metallography

Metallographic observations were performed by Olympus DSX1000 (Olympus Corporation, Shinjuku, Japan) optical microscope (OM). Quantitative image analysis was performed with ImageJ software (version 1.51j8, National Institutes of Health, Bethesda, MD, USA) on at least 20 micrographs for each material to evaluate the SiCp fraction. Assuming that the area percentage of SiCp in the image is equal to the volume percentage, then the weight percentage of SiCp in the matrix could be calculated by the volume percentage value. The ImageJ software was used for quantitative image analysis of SiC particles. The centre of mass of the SiC particles were then used as input data to calculate the first-, second-, and third-nearest neighbour distances (1NND, 2NND, and 3NND) using the MATLAB function ‘knnsearch’. A schematic representation of the ‘knnsearch’ function is depicted in Figure A2.

### 2.3. Mechanical Testing

Thermal compression tests were used to review the formability of composite materials in this research. Compression tests were carried out with a Zwick Roell Z100 (Zwick Roell, Ulm, Germany) testing unit at different strain rates: 0.001/s, 0.01/s, 0.1/s, and 1/s; and different temperatures: 25 °C, 350 °C, 420 °C, 470 °C. The current maximum operating temperature is 420 °C and the aim is to increase it to 470 °C. For this reason, the 350–470 °C temperature range was selected. The room temperature data are used as a reference for comparison. Before each test, the sample was heated for 10 min at the desired temperature. A compliance curve was registered to consider the stiffness of the machine in data analysis. The compliance curve was fitted with a linear function, and the related strain was removed from the curves.

## 3. Results

### 3.1. Microstructural Analysis

Figure 1 shows the microstructure of the five materials. The weight fraction of SiCp is listed in Table 2. The targeted quantity of 20 wt.% of particles was added to all the materials, but the transfer efficiency was not constant and resulted in different incorporated fractions depending on the size of the particles.

For the C0 matrix alloy, the transfer efficiency varied with the size of the SiC particles and the actual fractions were: 14% in the material C0_23 with 23 µm sized particles (Figure 1a), 19% for the 50 µm sized particles in material C0_50 (Figure 1b), only 4% in the material C0_10 with 10 µm sized particles (Figure 1c), and 10% for the mixture of 10, 23, and 50 µm sized particles, added at the same mass ratio, in the material C0_mix (Figure 1d). The material C1_23 also resulted in 14% of the 23 µm sized particles (Figure 1e). These results suggest that the particle size is the dominant parameter for transfer efficiency, almost 100% for the largest size of 50 µm and decreased to 20% for the smallest size of 10 µm.

Figure 1a shows the material reinforced with 23 µm sized SiCp, the phases are primary α-Al, and the binary Al–Si eutectic; the dispersion of SiCp particles is uniform, with almost no clusters observed. Figure 1b shows the material reinforced with 50 µm sized particles, the largest size in all materials, and Figure 1c shows the material reinforced with 10 µm sized particles. Due to the small size of the particles, a significant number of clusters were observed, and homogenous dispersion of particles was not achieved. This is visible in the significant standard deviation of the higher-order measures 2NND and 3NND listed in Table 3. Several particle-free areas confirm that the transfer efficiency was lower than other materials. Figure 1d shows the material reinforced with the mixed-sizes particles. The dispersion is better than in Figure 1c, and the addition of 50 µm size particles was limited, while most of the particles were 23 µm and 10 µm sized. There are several small clusters made of mixed-size SiC particles.

Figure 1e shows the 23 µm size SiCp reinforced Al matrix composite with RE addition. The particle dispersion was uniform, and the addition of La and Ce to the matrix alloy did not change the transfer efficiency of SiC particles; hence the SiC fraction of the C0_23 and C1_23 alloy was similar. The generation of new phases in the C1_23 material, previously identified as ɑ-Al_15_(Fe,Mn)_3_Si_2_, Al_20_(La,Ce)_3_Ti_2_ and Al_11_(La,Ce)_3_ [22], did not alter particle addition and distribution.

### 3.2. Mechanical Testing

Flow stress is described as a function of composition SiCp addition, temperature, strain, and strain rate. The true stress–true strain plots are given in Figure 2.

Figure 2a shows the temperature effect on the flow stress of material C0_23. The stress decreased as the compression temperature increased. The peak flow stress at room temperature reached a value of 426 MPa and was more than three times higher than the 350 °C peak stress value of 121 MPa. The 470 °C peak stress value of only 37 Mpa suggests that the softening mechanism is more powerful than the work-hardening mechanism at room temperature.

Figure 2b shows the strain rate effect on the flow stress of the C0_23 material at 420 °C and different strain rates. With the increase in strain rate, the peak flow stress of the curve increased. At the strain rates of 0.01 s^−1^, 0.1 s^−1^ and 1 s^−1^, the flow curve appeared smooth, while at the strain rate of 0.001 s^−1^, the curve appeared corrugated due to internal friction in the materials.

Figure 2c shows the true stress–true strain curve with different particle sizes and weight fractions at a constant temperature of 350 °C and a strain rate of 0.1 s^−1^ and C0 matrix. The peak value is at the range of 117–128 MPa, and the weight fraction is at the range of 4–19%. Even though the material C0_50 possesses 19 wt.% SiCp particle addition and 50 µm size particles reinforced, the peak stress is only 3 MPa higher than the C0_mix material. The lowest peak stress curve is the material C0_10, which displayed a peak stress of 117 MPa, 5 MPa lower than the material C0_23 and 11 MPa lower than the material C0_50. This result indicates that weight percentage and particle size slightly influence the flow stress during deformation.

Figure 2d shows the true stress–true strain curve of materials C0_23 and C1_23 at temperatures 420 °C and 470 °C. The materials C0_23 and C1_23 have the same particle volume fraction, 14 wt.% and the same particle size, 23 µm. At the temperature of 420 °C, the peak stress values of materials C1_23 and C0_23 were 90 MPa and 49 MPa, respectively. The peak stress value of material C1_23 at 470 °C is slightly higher than the material C0_23 at 420 °C. The difference between material C0_23 and material C1_23 is shown in Table 1. The addition of RE and transition elements almost doubled the strength compared to the C0_23 material. Figure 2d clarifies that the matrix alloy was more critical than the particle size and particle weight fraction on the effect of SiCp reinforced composite material deformation.

### 3.3. Zener−Hollomon Analysis

Equation (1) collects the constitutive equations used to calculate the material constants in the softening segment after peak stress:(1)$$\dot{\varepsilon }=\{\begin{array}{c} A\cdot \sigma^{n1}\cdot e^{-Q_{A}/\mathrm{RT}}\;\;\;\;\;\;\;\;\;\;\;\;\;\;\;\;\;\;\;\;\;\;\;\;\;\;\;\;\;\alpha \sigma <0.8\\ A\cdot e^{\beta \sigma }\cdot e^{-Q_{A}/\mathrm{RT}}\;\;\;\;\;\;\;\;\;\;\;\;\;\;\;\;\;\;\;\;\;\;\;\;\;\;\;\;\alpha \sigma >1.2\\ A\cdot {\left[\sinh\left(\alpha \sigma \right)\right]}^{n2}\cdot e^{-Q_{A}/\mathrm{RT}}\;\;\;\;\;\;\;\;\;\;\;\mathrm{for}\;\mathrm{all}\;\sigma \end{array}$$

Here n1, n2, α = β/n1 [1/MPa], β [1/MPa], and A [1/s] are material constants independent of temperature. α was described by Jonas et al. [35] as the reciprocal stress at which the strain rate changes from power to exponential dependence on stress. σ [MPa] is the flow stress, $\dot{\varepsilon }$ [1/s] is the strain rate, and R = 8.314 J/K·mol is the universal gas constant. Q_A_ [kJ/mol] is the activation energy of deformation and comes from Equation (2):(2)$$Q_{A}=R\cdot {\left[\frac{\partial \ln\dot{\varepsilon }}{\partial \ln[\sinh\left(\alpha \sigma \right)]}\right]}_{T}\cdot {\left[\frac{\partial \ln[\sinh\left(\alpha \sigma \right)]}{\partial \left(1/T\right)}\right]}_{\dot{\varepsilon }}$$

In simpler words, the activation energy indicates the energy barriers to plastic deformation during the hot deformation of metallic materials. If flow stress increases with the increasing Q_A_ value at the same deformation temperature and strain rate, it is suggested that the materials with lower Q_A_ value can be deformed more easily with a lower force. Figure 3 depicts the graphical solution of the Zener−Hollomon model in Equations (1) and (2). The parameter n1 is the slope in the graph ln$\dot{\varepsilon }$ v. σ (Figure 3a), and the parameter β is the slope in the graph ln$\dot{\varepsilon }$ v. lnσ (Figure 3b). The parameter n2 is the slope in the graph ln$\dot{\varepsilon }$ v. ln[sinh(ασ)] in Figure 3c and corresponds to the first term in Equation (2). The second term in Equation (2) is the slope in the graph ln[sinh(ασ)] v. 1000/T, in Figure 3d.

The material constants A and n2 were determined from the ln(Z) v. ln[sinh(ασ)] plot. Table 4 lists the material constants and the activation energy calculated for the investigated materials. The activation energy for the hot deformation of composite ranges 301–584 kJ/mol, which is higher than the bulk self-diffusion of pure Al 142 kJ/mol.

The Origin software was used to model the relationship between the activation energy Q_A_ (kJ/mol), the RE addition to the matrix alloy, and the SiC particles’ microstructural parameters. The multiple linear regression resulted in Equations (3) and (4):(3)$$Y=395\times f_{\mathrm{SiC}}+6\times d_{\mathrm{SiC}}+1871\times \left[\mathrm{RE}\;\mathrm{wt}.\%\right]+265,{\;R}^{2}=0.38$$
(4)$$Y=23\times d_{\mathrm{SiC}}+1166\times \left[\mathrm{RE}\;\mathrm{wt}.\%\right]-21\times 1\mathrm{NND}+420,{\;R}^{2}=0.619$$
where Y is the activation energy Q_A_ [kJ/mol], f_SiC represents the SiCp fraction, d_SiC represents the average Feret diameter [µm] of the carbides, [RE wt.%] is the RE content in the matrix alloy, 1NND [µm] is the first-nearest neighbour distance between SiC particles. These results suggest that particle size and the interparticle spacings are better factors than the fractions of particles added, as indicated by the R^2^-factor.

The particle fraction, particle size, and nearest neighbour distance are independent parameters, depending on each other. Equation (3) uses the SiCp fraction (f_SiC), SiCp particle size (d_SiC), and RE addition ([RE wt.%]). The parameters in Equation (3) show that an increase in any of the factors causes an increase in the activation energy. The value R^2^ = 0.38 illustrated that the fit is poor and does not describe the effect well. In Equation (4), the SiCp fraction was replaced by the nearest neighbour distance (1NND). An increasing 1NND represents a reduced clustering behaviour of the particles to some extent. Replacing the fraction SiCp with the 1NND increased the fit value to R^2^ = 0.62. The fact that R^2^ increased suggests that it is not the mass fraction of particles that matters but rather their arrangement. The main conclusion is that bringing particles closer, increasing their size, and adding RE to form intermetallic phases to lock the matrix from moving increase the activation energy and stabilise the materials at elevated temperatures.

### 3.4. Processing Maps

The processing map consists of the superimposed map of power dissipation and an instability map. These are developed based on the Dynamic Materials Model [36]. The objective is to manufacture components with controlled microstructure and properties without macro or microstructure defects. Power dissipation is the percentage of energy converted into thermal and microstructure change. The factor that partitions power into these two forms is the strain rate sensitivity exponent. The strain rate sensitivity exponent, m value, was estimated from Equation (5). The processing map consists of the superimposed map of power dissipation and an instability map. These are developed based on the Dynamic Materials Model [36]. The objective is to manufacture components with controlled microstructure and properties without macro or microstructure defects. Power dissipation is the percentage of energy converted into thermal and microstructure change. The factor that partitions power into these two forms is the strain rate sensitivity exponent. The strain rate sensitivity exponent, m value, was estimated from Equation (5):(5)$$m={\frac{\partial \left(\ln\left(\sigma \right)\right)}{\partial (\ln\left(\dot{\varepsilon }\right))}|}_{\varepsilon }$$
where m denotes the strain rate sensitivity of the flow stress at a constant strain of ε. A dimensionless parameter called efficiency of power dissipation η was defined in Equation (6):(6)$$\eta =\frac{2m}{m+1}$$

A dimensionless parameter called instability criterion ξ is used to obtain the instability map, and it was defined according to Equation (7):(7)$$\xi \left(\dot{\varepsilon }\right)=\frac{\partial \ln\left(\frac{m}{m+1}\right)}{\partial \ln\dot{\varepsilon }}+m>0$$

Figure 4a–e show the processing map of all five materials. The contour numbers represent power dissipation efficiency, and the shaded domains indicate the regions of flow instability, with ξ < 0. The purpose of the hot-processing map is to guide the choice of the metal-forming parameters to avoid macro and microstructural defects in a repeatable manufacturing environment [29].

The material C0_23 represents a baseline in this study as the standard 23 µm size SiCp reinforced composite without RE-additions. The processing map for the material C0_23 is shown in Figure 4a, where the contour numbers represent power dissipation efficiency, and the shaded domains represent the regions of flow instability. The processing map at different strain levels is shown together with the associated stress–strain curves at some preselected levels of strains from the strain at peak stress up to a strain of 0.5. During the whole process, the processing map could guide the choice of the hot working parameters for the composites. At the peak strain processing map, the highest power dissipation efficiency value is 0.25, and the stable region separates two regions of instability. Instability is generated by a lower temperature or high temperature and high strain rates. Increasing the strain in the materials closes that gap between the instability regions, forcing the stable region towards lower strain rates and higher temperatures. At strains equal to 0.3 and above, the process map appears to stop changing and stabilise, and the highest efficiency is 0.23. This behaviour approximately coincides with reaching a planar portion of the stress–strain curves, where there is a weak tendency towards softening after the peak stress (Figure 4a).

The material C0_50 is similar to the material C0_23 but has an increased SiCp particle size. Figure 4b), where there is a large and dominant unstable region and efficient deformation is only possible at low strain rates, or high strain rates and high temperature. The processing maps appear to stabilise already at strain above 0.2. It should be noted that there is a tendency to tolerate higher strain rates at the highest temperatures. A broad zone also allows deformation at an energy dissipation efficiency above 0.21.

In Figure 4c, the effect of RE elements in the material C1_23 shows similarities to both material C0_23 and C0_50. Firstly, there is a separation between the two unstable regions at peak strain. These two regions join, and just as in the material C0_23, the pattern stabilises the stable deformation is pushed towards lower strain rates. It should be noted that the peak stress is significantly higher than for both material C0_23 and C0_50 and is followed by a significant softening. The SiCp particles do not deform during the deformation process, and the matrix material absorbs all deformation. The RE-additions significantly increase the strength of the matrix, as seen in the peak strength in Figure 2d. The difference to the C0_50 material that would have a softer matrix is that at the highest temperature, the stable region is pushed further down towards low strain rates and that the energy dissipation goes above 0.25 compared to 0.21 for the C0_50 material. This outcome suggests increased formability at the highest temperatures.

The material C0_10 has a soft matrix with a lower fraction of smaller SiCp particles that seemed more clustered. As expected, the peak stress is lower, and the processing map is similar to the C0_23 material (Figure 4a,d). The processing map for the material C0_10 shows a relatively wide gap between the unstable regions. The degree of softening after the peak strength exists but it is weak, just as for the material C0_23. The change in the processing maps is similar, and the two instability regions join and forces the stable regions towards lower strain rates at strains from 0.3 and up. There is no significant breakdown at the maximum strain investigated. The low fraction and small size of the SiCp particles suggest that the matrix deformation dominates this type of processing map.

The material C0_mix is a mixture of large and small SiCp particles targeted to better lock the movement of the material flow, and Figure 4e shows a similar effect to the one seen with the RE-addition in Figure 4c. The two instability regions are joined at peak stress but separated at 0.1 to join again at 0.2. Strains from 0.2 and up appear to be stable, coinciding with a steady-state behaviour in the stress–strain curve. It should also be noted that for all the materials with the C0 matrix base materials (Figure 4a,d,e), except for C0_50 (Figure 4b), the stable regions tolerate a higher strain rate at temperatures around 420–440 °C. Strengthening the matrix with RE-addition shows the same behaviour, but the stable region was forced to lower strain rates.

## 4. Discussion

### 4.1. Strengthening Mechanisms

In Al-based composite materials, the strengthening can be classified as (i) load bearing, (ii) Hall–Petch mechanism, (iii) Orowan strengthening, and (iv) the modulus mismatch [37]. In Al-based composite materials, the strengthening can be classified as (i) load bearing, (ii) Hall–Petch mechanism, (iii) Orowan strengthening, and (iv) the modulus mismatch [37]. In the present study, the load bearing and the modulus mismatch significantly improve composite strength. The SiC particles used in this research are micro-scales, and only the nano-scale particles contribute to the Orowan strengthening. Figure 1 shows that the secondary dendrite arm spacing does not change significantly, so the Hall–Petch mechanism does not vary significantly and may be lumped into the σ_0_-term, Equation (8). Equation (8) shows the relation used to calculate the total flow stress with the different strengthening contributions listed in Table 5:(8)$$\sigma_{\mathrm{tot}}=\sigma_{0}+{\Delta \sigma }_{\mathrm{LB}}+{\Delta \sigma }_{\mathrm{MM}}+{\Delta \sigma }_{os}$$

Equation (9) was used to calculate the material C0 matrix strength σ_0_ without the reinforcing SiC particles. Equation (10) describes the contribution from load bearing, and Equation (11) describes the strengthening contribution from the modulus mismatch effect [37]. Equation (12) describes the contribution of the Orowan strengthening [38].
(9)$$\sigma_{0}=\frac{\left(\sigma_{C0\_10}-C_{\mathrm{MM}}\times \sqrt{\frac{f_{\mathrm{SiC}}}{d_{\mathrm{SiC}}}}\right)}{\left(1+0.5\times f_{\mathrm{SiC}}\right)}$$
(10)$$\sigma_{0}=\frac{\left(\sigma_{C0\_10}-C_{\mathrm{MM}}\times \sqrt{\frac{f_{\mathrm{SiC}}}{d_{\mathrm{SiC}}}}\right)}{\left(1+0.5\times f_{\mathrm{SiC}}\right)}$$
(11)$${\Delta \sigma }_{\mathrm{MM}}\approx C_{\mathrm{MM}}\times \left(\sqrt{\frac{f_{\mathrm{SiC}}}{d_{\mathrm{SiC}}}\times \varepsilon }+\sqrt{\frac{f_{\mathrm{int}}}{d_{\mathrm{int}}}\times \varepsilon }\right)$$
(12)$${\Delta \sigma }_{\mathrm{OS}}=\left(0.538\times \frac{G\times b\times \sqrt{f_{\theta }}}{d_{\theta }}\right)\times \ln\left(\frac{d_{\theta }}{2\times b}\right)$$

In Table 5, G is the shear modulus [GPa], b is Burger’s vector, d [m] is the actual diameter of the precipitates, f_SiC and f_int are the volume fractions of SiC particles and intermetallic phases, d_SiC and d_int are the characteristic dimension of the particles and σ_0_ [MPa] is the strength of the matrix alloy. The results of materials C0_50 and C1_23 are representative and presented in Figure 5a. The average error value between the experimental data and the calculated data in the material C0_50 is 7.65%, considered acceptable. Whereas the error value in the material C1_23 is large enough for reconsideration.

The Thermo-Calc software was used to calculate the volume fraction of precipitates formed in the material C1_23 at test temperature, θ-Al_2_Cu phases were predicted. The peak fraction of θ-Al_2_Cu was predicted to be 1.08% at 170 °C. This value was used to calibrate the Orowan contribution at 350 °C at $\dot{\varepsilon }$ = 1/s. The Orowan contribution at all other conditions was expressed as a function of particle size only. Figure 5b shows that the Feret diameter of θ-Al_2_Cu particle tends towards smaller sizes, as the deformation temperature increases, and test duration increases (i.e., reduced strain rate). Equation (12) was used in the material C1_23 to calculate strength contribution from precipitated θ-Al_2_Cu phases. The Orowan strengthening was fitted to match the gap between experimental data and calculated data for the material C1_23. The comparison of experimental data and calculated data confirmed that in the C0 matrix alloys, the strengthening mechanism was load bearing and modulus mismatch. In the C1_23 material, in addition to the above strengthening mechanisms, the Orowan strengthening should be included. The numerical results are presented in Table A1 and Table A2, Appendix B.

### 4.2. The Activation Energy

Equations (3) and (4) illustrate the relationship between the activation energy Q_A_ (kJ/mol), the RE addition ([RE wt.%]), and the main microstructural features of the SiC particles: the fraction (f_SiC), the average size (d_SiC), and the first-nearest neighbour distance (1NND). Figure 5 illustrates that the SiCp fraction on the *x*-axis is not sufficient to describe the evolution of the activation energy: the maximum value of Q_A_ does not correspond to the highest fraction of SiCp. The SiCp size and level of clustering determine the characteristic interaction distance to activate the material flow.

The Stoke–Einstein Equation (13) relates diffusion to viscosity [39,40]:(13)$$D=\frac{\mathrm{kT}}{6\pi r\times \mu }$$
where D [m^2^/s] is the diffusion coefficient, k = 1.381 × 10^−23^ J/K is the Boltzmann’s constant, T [K] is the temperature, r [m] is the radius of the particle, and µ [Pa*s] is the viscosity. Inserting the Arrhenius equations for D and µ in Equation (13) results in Equation (14):(14)$$D_{0}\exp\left(-\frac{Q_{D}}{\mathrm{RT}}\right)=\frac{\mathrm{kT}}{6\pi r\times \mu_{0}\exp\left(-\frac{Q_{A}}{\mathrm{RT}}\right)}$$
where Q_D_ [J/mol] is the activation energy for self-diffusion and Q_A_ [J/mol] is the activation energy for the material flow. D_0_ = 3.5 × 10^−6^ m^2^/s for aluminium. Rearranging and applying the natural logarithm results in Equation (15):(15)$$\frac{Q_{D}+Q_{A}}{\mathrm{RT}}=\ln\left(\frac{6\pi r\times \mu_{0}D_{0}}{\mathrm{kT}}\right)$$

Taking the ratio between two materials i and j and assuming a value for µ_0_, it is possible to solve for r_i in Equation (16) with the constant C in Equation (17) under the assumption of an initial size r_j:(16)$$r_{i}=\exp\left(\left(\ln\left(r_{j}\right)+C\right)\frac{{\left(Q_{D}+Q_{A}\right)}_{i}}{{\left(Q_{D}+Q_{A}\right)}_{j}}-C\right)$$
(17)$$C=\ln\left(\frac{6\pi \times \mu_{0}D_{0}}{\mathrm{kT}}\right)$$

In the C0_10 material, characterised by the smallest SiC size, the presence of clustering (Figure 1d) and the lowest SiC fraction, the characteristic size was assumed in the order of the atomic radius, 1.43 × 10^−10^ m or 1.43 Å, because the material flow mainly involved the matrix alloy. This material was taken as the reference one (material j in Equation (16)) for the other materials having higher SiC fractions, larger SiC sizes, and less frequent clustering. The µ_0_ value in Equation (16) was assumed, as in Equation (18), based on the von Mises flow stress criterion:(18)$$\mu_{0}=\frac{\sigma_{470\;{}^{\circ}C}}{\sqrt{3}\times {\dot{\varepsilon }}_{\max}}=\frac{38270000\;\mathrm{Pa}}{\sqrt{3}\times 1{\;s}^{-1}}=2.210\;\mathrm{Pa}\times s$$

In the C0_23 material, the characteristic interaction distance increased to 10 nanometres and even more to 100 microns for the C0_50 material. This outcome aligns with the increasing SiC fraction and size through materials C0_23 and C0_50 in Figure 6. The interaction changes from the atomic level to the level of the secondary phase and finally to the SiC particles’ order of magnitude.

The size mixture in the C0_mix material led to a complex interconnection between particles with different sizes and more frequent clustering events. This outcome determines the characteristic distance in the order of 1 cm to activate the material flow, which is the dimension of the sample. The entire structure is involved in the deformation, and thus the energy required to activate the material flow is 20 to 90% higher than the materials reinforced with one-sized SiC particles.

The role of RE-based phases can be observed by comparing the materials C0_23 and C1_23. The presence of Al_11_(Ce,La)_3_ and Al_20_(Ce,La)Ti_2_ phases and the Al matrix strengthened by Cu in solid solution constitute an additional obstacle to hot deformation, and this contributes to higher values of activation energy compared to the C0_23 material having the same fraction and size of SiC particles. The entity to be moved in the C1_23 material is in the micrometre range, two orders larger than the one in the C0_23 material.

The interaction distance affects the activation energy and not the SiCp fraction itself, and this result agrees with Equations (3) and (4) results with an improved R^2^-value for size and distance and not for fraction SiCp.

### 4.3. The Processing Maps

The processing maps provided the forming parameters to avoid defects during deformation. According to the processing maps, at strains above 0.3, the stable and unstable regions tended to stabilise and did not change significantly with additional straining. The highest dissipation efficiency values and the stable region were located at low strain rates, in the range 0.01–0.001/s. Below a strain of 0.3, the highest dissipation efficiency values and the stable region resulted in a high temperature range, 380–470 °C and 0.001/s strain rate. The previous studies on processing maps of Al-based composites did not compare the result at different strain levels. Hao et al. [27] investigated an Al-Cu/SiCp35 material and reported the processing map at the strain of 0.5, and it had a wide safe region, from 350 to 500 °C and 0.1–10/s strain rates. This result is very different from what was observed in the present study for the different materials: low strain rates, in the range 0.001–0.01/s, facilitate a stable forming operation of the composites. A similar result was reported by Xiao et al. [31], who investigated an Al-Cu/SiCp15 composite. Going from 0.3 to 0.5 strain, the unstable area expanded in the 1–10/s strain rate range at all temperatures. A comparison with the results from Huang et al. [30] and Ramanathan et al. [32] is not directly possible because the authors used the decimal logarithm of the strain rate instead of the natural logarithm to build the processing maps.

The processing map of the material C1_23 shows the safe region during whole strain located at a narrow area in the temperature range 420–470 °C and at 0.001 strain rate. This feature highlights that the RE addition limited the forming performance. The formation of stable phases and the reduced interaction distance of particles increased the activation energy and the difficulty of formability; this phenomenon locked the soft matrix, hindering the deformation of the material. On the other hand, the addition of La and Ce to the matrix alloy determines a doubled peak stress, and this is due to the presence of hard phases dispersed in the matrix. As previously reported [21,22], the Al_11_(Ce,La)_3_ and Al_20_(Ce,La)Ti_2_ phases strengthen the matrix alloy significantly, giving 15% higher elastic modulus and 55% higher strength at 300 °C. The material can be considered a two-level composite: both the (La,Ce)-based phases and the SiC particles act as reinforcement. A similar phenomenon occurs also in the C0_mix material. The large-sized and the small-sized particles size combined in an inter-locking structure that hindered the flow of the soft matrix and thus increased activation energy. This behaviour resulted in a limited stable domain, at 440–470 °C and a strain rate of 0.01–0.001/s.

## 5. Conclusions

The present study focuses on the effect of the addition of RE and different sizes and amounts of SiC particles on activation energy, processing map, and strengthening mechanism. Thermal compression tests were used to review the formability of composite materials and combined with microstructural analysis.

The dominant contribution to peak strength was the soft alloy matrix. The addition of RE and transition elements significantly impacted the peak strength through an interaction between the SiC particle and the RE-containing intermetallic compounds. The main reinforcing effects in the C0 materials were the load bearing and modulus mismatch strengthening, while the Orowan strengthening also played an essential role in the C1 composite due to the Cu addition. The safe-forming region in the RE-added composite was stable from peak stress to 0.5 strain in the temperature range 420–470 °C and at 0.001/s strain rate. The power dissipation was 0.23–0.28. For the C0 materials, the processing map tended to stabilise after 0.2 strain, with no further changes in the stable and unstable regions. The particle–particle interaction distance plays a central role in thermal stability. The diffusion–viscosity simile revealed that the scale of particle–particle interaction distance impacted the thermal stability. The impact revealed itself as a difference between activation energies, from the one of self-diffusion to the Zener–Hollomon one assessed from the visco–plastic deformation. The SiC particle size, the presence of thermal stable phases, and the SiC particle distance—all these parameters influence the interaction distance.

The use of Al-MMCs brake discs is one way to tackle emissions reduction, both weight-related and material-related emissions. Awe [8] highlighted that the automotive vehicle exhaust emissions reduced drastically from 2000 to 2014. The Al–Si/SiCp composite brake disk reduces exhaust emissions by being 50% lighter than the cast iron equivalent, and it also has higher wear resistance, uniform friction, light weight, and reduced light distance braking ability.

## Figures and Tables

**Figure 1 materials-15-03789-f001:**
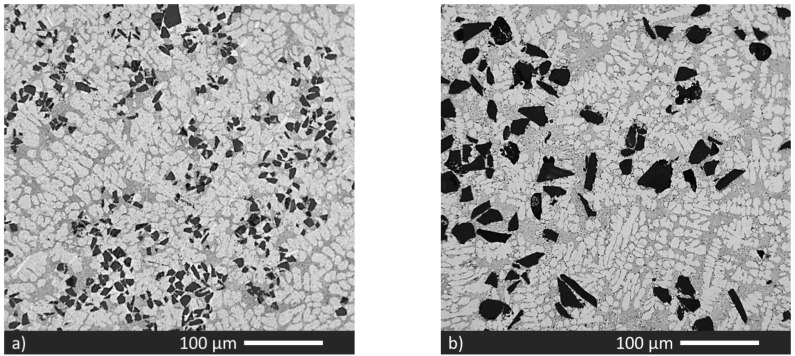
Micrographs: (**a**) material C0_23; (**b**) material C0_50; (**c**) material C0_10; (**d**) material C0_mix; (**e**) material C1_23.

**Figure 2 materials-15-03789-f002:**
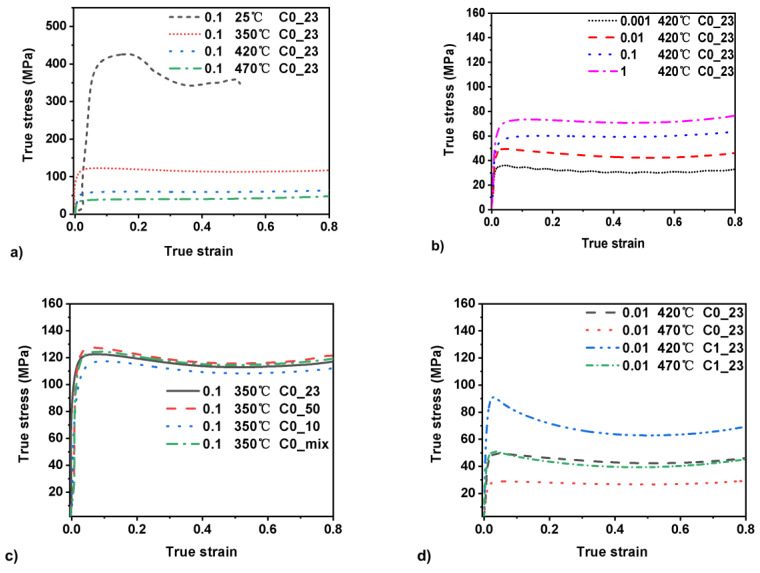
Compression curves: (**a**) material C0_23, the effect of the temperature at same strain rate; (**b**) material C0_23, the effect of strain rate at the same temperature; (**c**) constant temperature and strain rate, the effect of varying fraction; (**d**) materials C0_23 and C1_23 at constant strain rate, comparison at two temperatures.

**Figure 3 materials-15-03789-f003:**
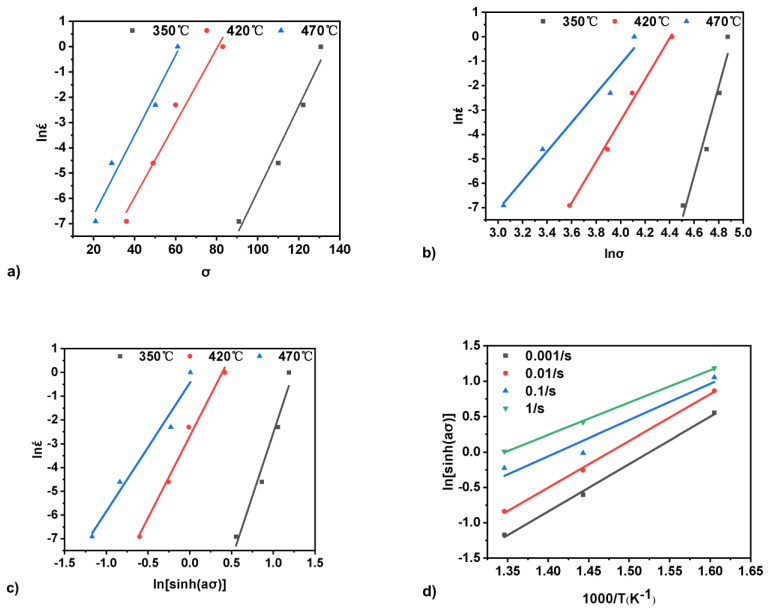
Graphical solutions for the Zener−Hollomon model: (**a**) the parameter n1 is the slope in the graph ln$\dot{\varepsilon }$ v. σ; (**b**) the parameter β is the slope in the graph ln$\dot{\varepsilon }$ v. lnσ; (**c**) the parameter n2 is the slope in the graph ln$\dot{\varepsilon }$ v. ln[sinh(ασ)]; (**d**) the second term in Equation (3) is the slope in the graph ln[sinh(ασ)] v. 1000/T.

**Figure 4 materials-15-03789-f004:**
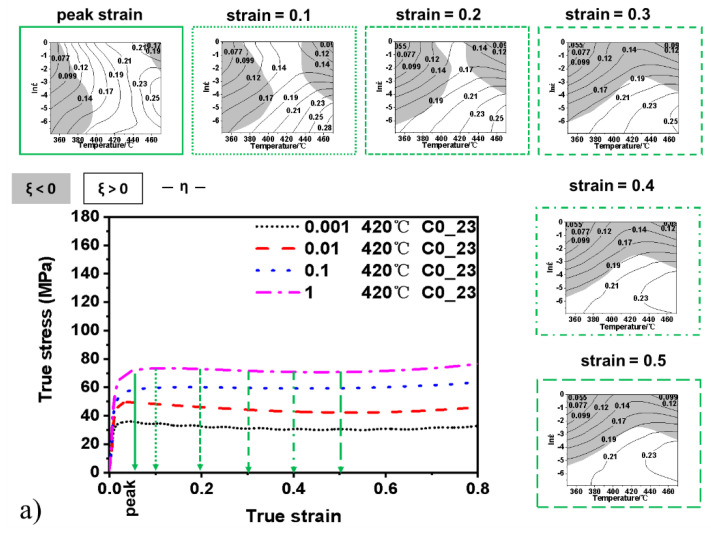
Processing maps of the investigated materials at different strain levels and related compression curves at 420 °C and different strain rates: (**a**) material C0_23; (**b**) material C0_50; (**c**) material C1_23; (**d**) material C0_10; (**e**) material C0_mix.

**Figure 5 materials-15-03789-f005:**
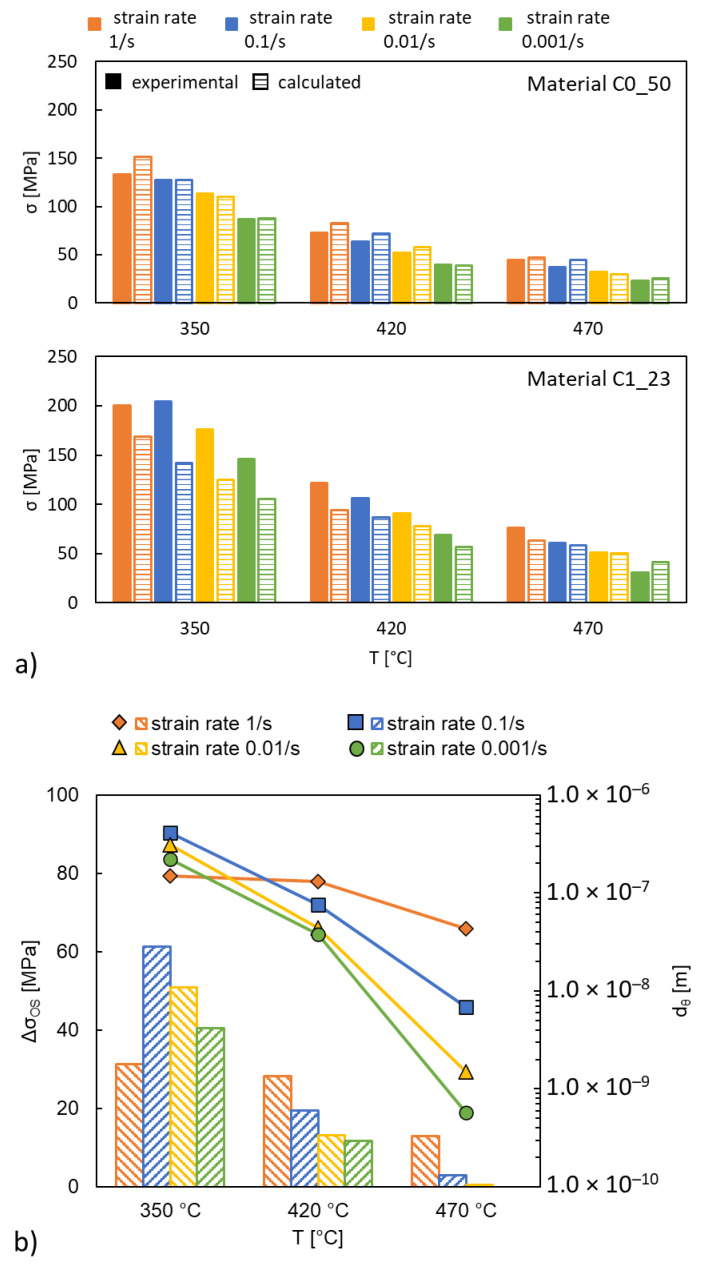
(**a**) The comparison between experimental data and calculated values with Δσ_LB_ and Δσ_MM_ for materials C0_50 and C1_23; (**b**) fitted Feret diameter d_θ_ of θ-Al_2_Cu particles and the resulting Orowan strength contribution Δσ_OS_.

**Figure 6 materials-15-03789-f006:**
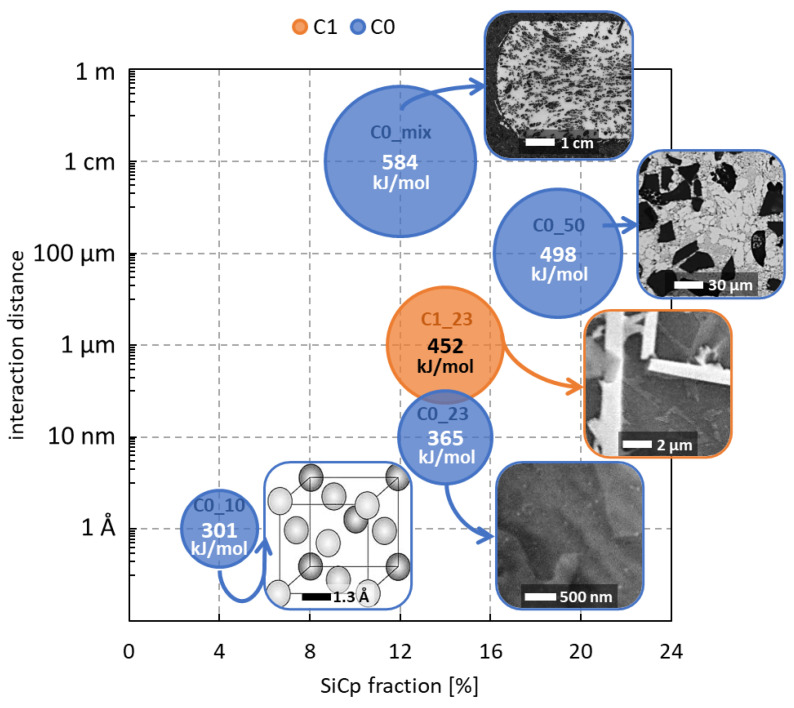
Relation between the activation energy Q_A_, the SiCp fraction and the interaction distance. The size of the bubble is proportional to the Q_A_ value.

**Table 1 materials-15-03789-t001:** List of automotive application of Al-based MMCs with different reinforcements.

Matrix	Reinforcement	Applications	References
Al and Al alloys	Al_2_O_3_	Piston rings, connecting rods	[10,11]
	Al_2_O_3_-Cf	Engine blocks	[12,13]
	MoS_2_p	Drive shafts	[14]
	B_2_O_3_w	Piston rings	[12,13]
	SiCp	Brake rotors, pistons, propeller shafts	[15]
	SiCw	Connecting rods	[16]
	TiCp	Pistons, connecting rods	[17]

f: fibre; p: particle; w: whisker.

**Table 2 materials-15-03789-t002:** Chemical composition [wt.%] of the composite materials and the weight fraction of SiCp.

Matrix Alloy	Si	Cu	Ni	Fe	Mn	Ti	Mg	Ce	La	Al	Code Name	SiC Size [µm]
C0	10	0.2	-	0.1	-	0.1	0.8	-	-	bal.	C0_23	23
											C0_50	50
											C0_10	10
											C0_mix	23 + 50 + 10 ^1^
C1	10	1.9	1.9	0.1	0.8	0.3	0.8	1	1	bal.	C1_23	23

^1^ Different sizes were added at the same mass ratio in material C0_mix.

**Table 3 materials-15-03789-t003:** Results of the quantitative image analysis performed using the ImageJ and MATLAB software. SD = standard deviation.

Material	SiC Fraction ± SD [wt.%]	SiC Size ± SD [µm]	1NND ± SD [µm]	2NND ± SD [µm]	3NND ± SD [µm]
C0_23	14 ± 2.7%	14 ± 0.9	14 ± 0.7	19 ± 1.1	24 ± 1.5
C0_50	19 ± 2.1%	32 ± 1.4	30 ± 2.5	41 ± 3.1	52 ± 3.3
C1_23	14 ± 1.9%	15 ± 1.1	16 ± 0.8	22 ± 1.2	28 ± 1.6
C0_10	4 ± 1.6%	12 ± 1.8	19 ± 4.7	32 ± 14.8	41 ± 17.1
C0_mix	12 ± 3.3%	19 ± 1.5	18 ± 1.8	26 ± 2.7	33 ± 3.6

**Table 4 materials-15-03789-t004:** Material constants and activation energy of the hot compressed composites, evaluated from the constitutive relations in Equation (2).

Material	n1	β [1/MPa]	α [1/MPa]	Q_A_ [kJ/mol]	n2
C0_23	10.9	0.158	0.0145	365	7.63
C0_50	12.2	0.224	0.0184	498	8.45
C1_23	11.8	0.126	0.0107	452	8.10
C0_10	8.62	0.146	0.0170	301	6.00
C0_mix	13.6	0.254	0.0187	584	9.07

**Table 5 materials-15-03789-t005:** Equations used for the contribution of the different strengthening mechanisms.

Strengthening Mechanism	Relation	Values
Matrix alloy	Equation (9)	C_MM_ = 1.47 MPa·√m [21]
Load bearing [37]	Equation (10)	
Modulus mismatch [37]	Equation (11)	C_MM_ = 1.47 MPa·√m [21]
Orowan [38]	Equation (12)	G = 41 GPa [22] b = 0.286 nm

## Data Availability

Data available in a publicly accessible repository.

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
