# Peer review of "The Influence of Ce, La, and SiC Particles Addition on the Formability of an Al-Si-Cu-Mg-Fe SiCp-MMC"

_materials, 2022, doi:10.3390/ma15113789_

Round 1
Reviewer 1 Report
This paper investigates the the formability of Al-Si alloys reinforced with different fractions and different sizes of SiCp to create an efficient Al-MMC brake disk. This topic is quite interesting. However, it needs some room of improvement to enhance the quality of the paper. Thus, I recommend the minor revision before it can be considered to be published.
Several aspects need to be improved:
- For abstract, please add and summarized all the literature findings with the significant results such as metallography and material properties. In addition, please write the aim of this research article to visualize the significant of this work for readers. Moreover, it needs to write a compact conclusive statement for this research paper article in order to indicate the potential of Al-MMC for brake disc applications.
- The introduction section is discussing on the backgrounds with current literature surveys. It should be noted that normally 'Introduction' should give comprehensive background through literature survey for the previous study by citing previous published work where-by scientific gaps that exist should be brought out. Thus, it is suggested to rewritten this section and each paragraph represent: 1) general issues/information; 2) Al-MMC background; 3) methods of analysis; 4) current research gaps; 5) potential of the metallic filler in automotive applications.
- Moreover, the amount of references is less in number and have to be added. It should be note that the number of references for research article should be more than 50. It is suggested to detail out the background of the recent studies on general Al-MMC and other additives uses for composites in many applications, its potentials by referring and citing these relevant articles such: 3390/polym14071367 and 10.3390/polym13111701.
- Please add a summarization table of previous findings on Al-MMC and other additives in metal composites for many applications including automobile applications in Introduction section. Please ensure it should be more than 7 references in the table.
- For Subtopic 3, please add a schematic diagram which explain the flow of the methods you conduct in the experiment. This is important to help the reader to understand the work that have you done.
- For Subtopic 4, all results you mention in the analyses are only is statement. Justify all the statements given in conjunction to other published literatures. In this case, a comprehensive explanation with your results are required to ensure the quality of the manuscript are valid and repeatable by other researchers. Please combine the discussion (Subtopic 5) and results (Subtopic 4) in order to avoid a hanging discussion analysis.
- For each sections, please add discussion which discussed from previous literatures/findings. In this literature discussions, it is essential to include the key elements such as ‘what is the research?’, methods (material and fabrications), significant findings, and correlate with other literature findings to support the significant of the literature results.
- Please move Tables 5 and 6 to appendix. Additionally, specifically for Table 6, please replace it by making an analytical figure.
- Please add one section to explain future outlook of Al-MMC in metal composites for brake disc applications in automotive industry. In this section, please add figures and tables to indicate the Al-MMC is significant values for this sector.
- For conclusion part, it is do not reflect what had been achieved including many speculations. It is too long and should be in one paragraph. Hence these need to be suitably modified. It may be remembered that this Section forms a summary of all the major observations/ results obtained. Accordingly, here presentation should consist of the main numerical results or the observations of the study briefly. Moreover, the authors have to include and highlight also the objectives and novelty of the work. This section also needs to explain on the research gap produce from the research. Indicate also the future work may conducted in the study. Hence better to rewrite this section based on the comments given in the whole text.
- Need to add more literatures in the research manuscript since only 20 references cited which is very less. In general, a good research paper should have more than 50 references in order to justify the state-of-art for the research study.
- Throughout this paper, there is need for better language throughout the manuscript. Please check with English native speaker to improve the readability of the review paper.
- Generally, the paper though contains some interesting results and novelty work, it lacks in its proper presentation in the whole manuscript. In view of these, the paper is highly recommended and should be accepted for publication in the revised form. It is suggested that the authors should revise the paper in the light of above comments/suggestions.

Reviewer 2 Report
Title: The influence of Ce, La, and SiC particles addition on the formability of an Al-Si-Cu-Mg-Fe SiCp-MMC
Manuscript ID: materials-1723230
Reviewer Comments: After looking at the manuscript, I would suggest that the author responds to these suggestions made:
- Make a list of the abbreviation and notation.
- Table 1 represents the chemical composition [wt.%] of the composite materials. With the equipment you have carried out, write a few lines about it.
- The author stated the MATLAB function used in line no. 109. Include a flow chart that shows how you used input and output values to do the analysis.
- Add a graphical abstract to the presented work to attract the researchers.
- In the introduction, add a section related to the Zener-Hollomon model.
- The conclusion must include the outcome. The fifth point in the conclusion must be revised.
- Add a methodology flow chart to your work.
- From 13 to 18, prepare the table and write the significant of all the equations and from where you have taken the equation (reference).
- For Table 6, it is better to plot the graph. Table 5 shows the comparison of experimental data and calculated values. For this table, write a paragraph about how its two parameters will help composite materials become stronger and more stable.
- The discussion section is very weak. Add more analysis and support with references.
- In lines 267–270, justify the statement.
- You have not supported your results with experiments and you have carried out MATLAB analysis. It makes it difficult to understand the behaviour of composite materials.
- Include the physical properties of composite materials in their procurement stage.
- Overall, check the language of the paper and verify the results.
- Rewrite the induction with reference to the latest papers cited. include more analysis and support the same with references for section 3.1., 3.2 and 3.3 5, In section 4.2, add these papers to support your results.
doi.org/10.1016/j.compstruct.2022.115242
doi.org/10.1016/j.compstruct.2022.115199
doi.org/10.3390/polym14071421
Reviewer 3 Report
The issues addressed in the publication are extremely topical in today's world. On the one hand, the use of composites reduces weight, which in case of application in motor vehicles reduces fuel consumption, as well as the amount of harmful substances emitted into the atmosphere.
The title of the publication was correctly formulated.
The content of the publication has been presented in a clear way for the reader, meeting the requirements of a scientific publication.
Tables and figures are clear.
The language used is correct.
The references included in the publication are sufficient.
I found only a minor editorial error. The authors in their publication often put a space between the value and the designation of degrees Celsius.
With the corrections made, I recommend the paper for publication.
Reviewer 4 Report
The influence of Ce, La, and SiC particles addition on the formability of an Al-Si-Cu-Mg-Fe SiCp-MMC
In this manuscript, authors reported the addition of particles materials to increase strength. I recommend major revision before publishing in the Journal of Materials.
Major revision
- The concurrency of abstract and manuscript is not related.
- The particle size and micro-structural analysis of the added particles should be included.
- The XRD analysis should be included for all the composition to identify the micro-structural change.
- Thermal analysis should be included in revised version (TGA). Is there any specific reason for the limited temperature analysis?
